# The Use of PI-FAB Score in Evaluating mpMRI After Focal Ablation of Prostate Cancer: Is It Reliable? Inter-Reader Agreement in a Tertiary Care Referral University Hospital

**DOI:** 10.3390/cancers17061031

**Published:** 2025-03-20

**Authors:** Elena Bertelli, Michele Vizzi, Martina Legato, Rossella Nicoletti, Sebastiano Paolucci, Ron Ruzga, Simona Giovannelli, Francesco Sessa, Sergio Serni, Lorenzo Masieri, Riccardo Campi, Emanuele Neri, Simone Agostini, Vittorio Miele

**Affiliations:** 1Department of Radiology, Careggi University Hospital, Largo Brambilla 3, 50134 Florence, Italy; michele.vizzi@unifi.it (M.V.); martina.legato@gmail.com (M.L.); ronruzga@gmail.com (R.R.); simona.giovannelli@unifi.it (S.G.); agostinis@aou-careggi.toscana.it (S.A.); mielev@aou-careggi.toscana.it (V.M.); 2Unit of Urological Minimally Invasive, Robotic Surgery and Kidney Transplantation, Careggi University Hospital, Largo Brambilla 3, 50134 Florence, Italy; rossella.nicoletti@unifi.it (R.N.); francesco_sessa@hotmail.it (F.S.); sergio.serni@unifi.it (S.S.); lorenzo.masieri@unifi.it (L.M.); riccardo.campi@unifi.it (R.C.); 3Department of Health Physics, Careggi University Hospital, Largo Brambilla 3, 50134 Florence, Italy; sebastiano.paolucci@unifi.it; 4Department of Experimental and Clinical Medicine, University of Florence, 50134 Florence, Italy; 5Academic Radiology, Department of Translational Research, University of Pisa, Via Roma, 67, 56126 Pisa, Italy; emanuele.neri@unipi.it

**Keywords:** PI-FAB, focal therapy, HIFU, prostate cancer, mpMRI

## Abstract

Focal therapies are very promising techniques for prostate cancer treatment, but their use in post-therapy MRI assessment, both in image acquisition and reporting, requires standardization. Prostate Imaging after Focal Ablation (PI-FAB) scoring, first described in May 2023, is a new scoring system for interpreting prostate multiparametric magnetic resonance imaging (mpMRI) results after focal therapies. The aim of our study was to assess the inter-reader agreement of PI-FAB scores among radiologists in a single large cohort of patients treated with focal therapy (in particular, high intensity focused ultrasound, HIFU) in a tertiary care referral University Hospital. To our knowledge, only two studies have evaluated the inter-reader agreement of PI-FAB scoring so far. In our experience, this new score significantly improves the inter-reader agreement of radiologists with different levels of experience. Furthermore, HIFU seems to be a very promising tool for the treatment of selected prostate cancer patients.

## 1. Introduction

Prostate cancer (PCa) is currently the second most common cancer in men in terms of incidence, with approximately 1.4 million new cases reported in 2022 [1]. Although in recent years there has been a reduction in or stabilization of mortality rates, the burden of disease remains substantial [2].

In the context of localized disease, focal therapy (FT) is now a promising alternative to radical treatments in selected patients, intending to evade the potential side effects associated with the other available active treatments, namely surgery and radiotherapy [3,4,5]. In fact, radical prostatectomy (RP) has an immediate impact on sexual function, with limited recovery and persistent symptoms, as well as urinary leakage, impotence, and rectal problems that affect quality of life [6,7]. Ten different FT modalities have been described: high intensity focused ultrasound (HIFU), focal cryotherapy, irreversible electroporation (IRE), focal brachytherapy, focal laser ablation (FLA), photodynamic therapy (PDT), microwave ablation, partial prostatectomy, bipolar radio frequency ablation (bRFA), and prostatic artery embolization (PAE) [4]. Although FT is a promising alternative treatment for localized PCa in terms of both oncological and functional outcomes [8,9,10] with an acceptable rate of adverse events [11], there is an unmet need for validation of non-invasive methods to evaluate treatment success or failure and, therefore, improve follow-up pathways after FT [12,13].

Recently, Giganti et al. proposed the Prostate Imaging after Focal Ablation (PI-FAB) scoring system [14], a new MRI-based standardized system to evaluate post-FT MRI results. The score is based on the features of other well-established scoring systems used in PCa multiparametric magnetic resonance imaging (mpMRI), like the prostate imaging reporting and data system (PI-RADS) [15], as well as other new systems like prostate imaging quality (PI-QUAL) [16], prostate imaging for recurrence reporting (PI-RR) [17], and prostate cancer radiological estimation of change in sequential evaluation (PRECISE) [18,19]. It employs a three-point scale to assess three MRI sequences in sequential order (DCE sequences, DWI starting with evaluation of the high-b-value sequence followed by the ADC map and T2). In the PI-FAB scoring system, it is advised to compare post-treatment mpMRI images with the pre-treatment ones, especially for scores 2 and 3 [14]. To our knowledge, only two studies have evaluated the inter-reader agreement of PI-FAB so far [20,21].

Our aim was to assess the inter-reader agreement of PI-FAB scores among three radiologists with varying levels of experience in a single large cohort of patients treated with HIFU in a tertiary care referral center.

## 2. Materials and Methods

### 2.1. Study Design and Patient Population

Between July 2019 and June 2024, of the 92 patients treated with HIFU, the most used focal therapy technique at our center, 68 were included in this single-center, retrospective, observational study for a total of 109 mpMRI scans. Patients without pre-treatment (n = 12) and post-treatment (n = 9) MRI results, as well as those for whom the DCE sequence was not utilized (n = 3), were excluded (Figure 1).

All patients underwent the first follow-up mpMRI 6 months after FT; 30 patients underwent another evaluation at 18 months and 11 had an additional evaluation at 30 months. All the mpMRIs, both before and after focal therapy, were acquired using the same protocol and reported in our radiology department.

Patients without a pre-treatment mpMRI, post-treatment MRI, or DCE (bi-parametric MRI) were excluded. All patients were surgery-naive before the FT and after the mpMRI they underwent a fusion biopsy to confirm the presence of PCa.

### 2.2. HIFU Treatment

The study protocol used to enrolled patients in focal therapy was approved by the local institutional ethics review board (CEAVC n° 21111 prot florence focal) and was conducted in accordance with the ethical standards of the Helsinki Declaration of 1975 and its later versions. Written informed consent was obtained from all patients prior to study enrolment.

Men aged > 18 years old who were diagnosed with PCa and adhered to the following criteria were considered potential candidates for HIFU: clinical tumor stage ≤ T2, visible index lesion(s) on multiparametric MRI less than 20 mm in diameter, PSA ≤ 10 ng/mL, ISUP grade ≤ 2. Patients were excluded in cases of previous radiotherapy treatment for prostate neoplasm.

HIFU therapy was performed by 2 surgeons using the Sonablate^®^ 500 device (Sonacare Inc., Charlotte, NC, USA). Treatments were delivered following the principles already available in the literature [22]: a focal lesion ablation or quadrant fashion was used depending on the gland volume, tumor volume, and its location; an ablation margin of 5 mm was adopted in the treatment. Treatments were performed under general anesthesia with real-time ultrasound monitoring. An indwelling catheter was inserted after the procedure and removed 7 days after.

### 2.3. mpMRI Protocol

All the mpMRI scans were acquired at our radiology department and conducted with an acquisition protocol aligned with that proposed by PI-RADS v.2.1 and described in detail in our previous work [23].

Both pre- and post-treatment scans were acquired using a 1.5 T MR scanner equipped with an 18-channel anterior pelvic phased-array coil and a 16-channel posterior spine phased-array coil (Magnetom Aera, Siemens Medical Systems, Erlangen, Germany).

The protocol included high-resolution T2-weighted turbo spin-echo (TSE) sequences in the axial, sagittal, and coronal planes (Slice Thickness 3 mm without gap; Matrix 272(P) × 320 (F); FOV (200 mm × 200 mm); a T1-weighted pre-contrast spin-echo (SE) sequence in the axial plane; a multi-b DWI (50, 500, 800, 1000 s/mm^2^) (EPI-DWI) sequence from which corresponding ADC maps were obtained; a multi-b DWI (1400–1800 s/mm^2^) (EPI-DWI) sequence; and a DCE assessment with fat suppression gradient-echo 3D T1W sequences with high time resolution (<7 s) [23].

### 2.4. mpMRI Image Analysis

All the mpMRIs were independently analyzed by three radiologists dedicated to urogenital imaging. The three readers had 12 years (500 prostate mpMRIs per year), 8 years (450 prostate mpMRIs per year), and 3 years (300 prostate mpMRIs per year) of experience, respectively. A PI-FAB score from 1 to 3 was assigned for each exam, with the following rules:

PI-FAB 1: No signs of enhancement within the site of the original tumor or an enhancing linear area not at the site of the original tumor or at the edge of the ablation cavity. Monitoring is recommended.

PI-FAB 2: A focal enhancing area ≤ 3 mm at the site of the original tumor. The need for biopsy is decided based on PSA kinetics.

PI-FAB 3: Early focal enhancement > 3 mm within the ablation zone/edge of the ablation cavity or a PI-FAB 2 focus showing increased size. Biopsy is recommended.

The sequential order of assessment of the various sequences proposed in the PI-FAB guidelines (DCE, DWI, ADC, T2WI) was followed by all readers.

The radiologists were blinded to all patient data (e.g., PSA kinetics, histopathological results, imaging reports, time elapsed since treatment), except for the area treated with focal therapy in the pre-treatment MRI scans. All scans were anonymized and presented randomly (not sequentially if related to the same patient).

### 2.5. Fusion Targeted Biopsy After Focal Therapy

The lesions described as suspicious were biopsied by an expert uroradiologist with more than 30 years of experience in prostate biopsies. The type of biopsy used was a free-hand transperineal MRI/US fusion-guided targeted biopsy with a virtual navigation platform (MyLabTM Twice Esaote, Genoa, Italy). The suspicious mpMRI lesions were sampled. For each lesion, 3–5 targeted cores were obtained in relation to their size. All patients also underwent a randomized standard prostate biopsy of 8–14 cores (6–10 samples for the peripheral zone and 2–4 samples for the transition zone) in relation to prostate volume.

The histological evaluation was conducted by an expert pathologist with more than 20 years of experience in prostate cancer using the WHO/ISUP grading system.

### 2.6. Statistical Analyses

To address the substantial class imbalance in the dataset, the inter-rater agreement of the PI-FAB scoring system among the three radiologists was assessed using Gwet’s AC2 statistic. The standard error (SE), 95% confidence interval (95% CI), and *p*-value (*p*) were also calculated. In particular, the *p*-value was extracted using a Wald test, which tests the null hypothesis that Gwet’s AC2 equals zero by comparing the ratio of the AC2 value to its SE against the standard normal distribution. A value of *p* < 0.05 was considered significant.

The interpretation of the agreement levels of Gwet’s AC2 coefficient was conducted according to Gwet’s guidelines: agreement was considered poor if the Gwet’s AC2 value was less than 0, slight between 0 and 0.20, fair between 0.21 and 0.40, moderate between 0.41 and 0.60, good between 0.61 and 0.80, and very good between 0.81 and 1.00 [24].

A value of *p* < 0.05 was considered significant. The statistical analysis was performed using R software (version 4.3.1). Specifically, the “irrCAC” package was used to calculate Gwet’s AC2 statistic [25].

## 3. Results

Prior to treatment, the mean age among the patients was 70.6 years ± 8.31 (range 52–86 years) with a mean PSA of 7.85 ± 1.21 ng/mL (range 7.2–9.8 ng/mL); at the first follow-up, the mean PSA was 4.64 ng/mL (range 3.9–8.2 ng/mL). The PI-RADS score assigned to the target lesion before treatment was 3 in 26 cases and 4 in 42 cases. The average tumor size in T2-weighted images was 7 mm (range 3–14 mm). A total of 25 (36.8%) lesions showed contrast enhancement, with a mean ADC value of 1025 mm^2^/s (range 743–1343). The histopathological result of the pre-treatment biopsy showed a WHO/ISUP grade 1 (Gleason score ≤ 6) in 61 patients and grade 2 (Gleason score 3 + 4) in 7 patients.

The patient characteristics are summarized in Table 1.

For the assessment of inter-reader agreement among the three radiologists, we evaluated the distribution of the scores assigned to all 109 mpMRI examinations. The distribution was as follows: the most experienced radiologist assigned 105 PI-FAB 1, 1 PI-FAB 2, and 3 PI-FAB 3 scores; the moderately experienced radiologist assigned 104 PI-FAB 1, 1 PI-FAB 2, and 4 PI-FAB 3 scores; the least experienced radiologist assigned 94 PI-FAB 1, 6 PI-FAB 2, and 9 PI-FAB 3 scores. During the retrospective analysis of the post-biopsy histopathological reports of the three patients with a PI-FAB score of 3 reported by the most experienced radiologist, they were all confirmed as PCa WHO/ISUP grade 2.

Gwet’s AC2 statistic was calculated to evaluate the inter-reader agreement for PI-FAB, which reflects how similarly the operators interpreted the scans. Gwet’s AC2 coefficient shows a value of 0.941 (SE = 0.019, 95% CI = [0.904–0.978], *p* < 0.0001), indicating a very high level of agreement among the three observers regarding the interpretations of the mpMRI scans.

In particular, there was complete agreement among the three radiologists for 93 scores of 1 and 3 scores of 3.

A heatmap of scores assigned by the readers, offering a visual representation that facilitates the comparison of the ratings, is illustrated in Figure 2.

Subsequently, as shown in Table 2, we independently assessed the results of the most experienced radiologist for each follow-up scan to determine how many MRI scans showed evidence of significant recurrence (Table 2).

Specifically, after 6 months, there were 64 (94.14%) PI-FAB 1 scores (Figure 3), 1 (1.47%) PI-FAB 2 score, and 3 (4.41%) PI-FAB 3 scores (Figure 4); at 18 months, there were 30 (100%) PI-FAB 1 scores, and at 30 months, 11 (100%) PI-FAB 1 scores.

## 4. Discussion

Before the introduction of the PI-FAB scoring system, some studies attempted to determine the likelihood of recurrence after focal therapy using the PI-RADS score, but with limited success.

Regarding the diagnostic performance of mpMRI in predicting PCa recurrence after HIFU, in a recent meta-analysis [26], the sensitivity and specificity were 0.81 (95% confidence interval [CI] 0.72–0.90) and 0.91 (95% CI 0.86–0.96), respectively. However, due to the significant heterogeneity among the included studies, intrinsic limitations were noted. Specifically, studies with a smaller patient population showed higher sensitivity and specificity in predicting recurrent PCa than those with a larger population. For instance, Mortezavi et al. reported a sensitivity of 14.3% in the diagnosis of clinically significant prostate cancer [27].

PI-RADS was developed for treatment-naive patients and it is difficult to detect some types of prostate tumors using mpMRI, for example those that are low-grade, those that are organ-confined, and the smaller ones [28].

PI-FAB is a scoring system introduced very recently, so only three studies have attempted to determine its diagnostic performance, with rather conflicting results. In the study by Pausch et al. the most experienced reader had sensitivity, specificity, positive predictive value (PPV), and negative predictive value (NPV) at 6 months after focal therapy of 43%, 97%, 60%, and 94%, respectively [21]. Gelikman et al. found that the most experienced reader had a sensitivity of 92.9%, specificity of 62.5%, PPV of 59.1%, and NPV of 93.8% [20].

Our results differ from those of Pausch et al. [21]: in their study, the results of reader 2 were mostly relatively low, with high disparities between the two readers, so the authors suggest the potential influence of the radiologists’ experience. Instead, in our study the use of PI-FAB seems to significantly reduce the differences created by the radiologists’ experience. Furthermore, they analyzed a relatively old patient series (the mpMRI were collected from April 2014 to April 2019). Hence, it is understood that the mpMRIs were acquired according to the PI-RADS 2 protocol and not the PI-RADS 2.1 protocol, which is currently in use. In fact, the authors underline the fact that some of the mpMRI results were acquired with the endorectal coil, which is no longer recommended. Gelikman et al. [20] also analyzed a case series collected over 10 years (from 2013 to 2023), which is very heterogeneous as, since this period, the versions of PI-RADS have changed, as have the optimal minimum technical requirements for the acquisition of MRI exams, as well as for their reporting. For example, in their series some of the mpMRI were also acquired with the endorectal coil. Instead, we analyzed a series of patient cases collected between July 2019 and June 2024, and all the mpMRI were acquired using PI-RADS version 2.1.

Furthermore, Gelikman et al. [20] examined a limited patient cohort (38 patients) receiving various types of focal therapy: focused laser ablation (FLA), HIFU, and cryoablation. Only FLA procedures were conducted at the authors’ institution, whereas HIFU and cryoablation patients were referred to their hospital for further workup due to suspicion of recurrence. In contrast, we investigated a significantly larger and more selectively defined population—nearly double the size—in which all patients underwent the same focal therapy (HIFU) at our institution. Additionally, in our work all the mpMRIs, both pre- and post-treatment, were conducted at the same institution using identical MRI scanners and protocol.

Although our study was not aimed at assessing diagnostic performance, one limitation is the absence of histological findings (fusion biopsy or surgery) for data correlation in all patients, due to the retrospective nature of the study and the high number of non-suspicious findings, which were reported as PI-FAB 1 and do not necessitate biopsy, but require subsequent follow-ups over time. On the other hand, this high percentage of PI-FAB 1 (94.1% at 6 months, 100% at 18 months and 30 months) suggests two possible assumptions: first, HIFU is a reliable technique with a very low rate of recurrence at the treatment site; second, if disease recurrence is present, it becomes evident from the first follow-up, as no lesions with a PI-FAB score greater than 1 were detected in subsequent evaluations. This, however, as previously mentioned, does not provide definitive evidence of its success due to the lack of biopsy confirmation.

Our study is currently the only one in the literature that assesses the inter-reader agreement among more than two radiologists. Notably, even if we increase the number of radiologists, the agreement among operators with different experience is excellent, with statistically significant values, in contrast with previous findings. In fact, both Gelikman and Pausch report an inter-reader agreement that is lower than ours. In our series, the best inter-reader agreement due to the use of PI-FAB scoring was observed among the two more experienced radiologists, while the less experienced one assigned a higher number of PI-FAB 2 scores (very low anyway).

One of the main limitations of the study is the pronounced class imbalance in the dataset, with the PI-FAB score of 1 being predominantly represented. It may partially explain the difference in agreement compared to the other two studies, as the findings observed in PI-FAB scores 2 and 3 are more challenging to interpret. In these scenarios, agreement tests for more than two observers, such as Fleiss’ kappa and Krippendorff’s alpha, may lose their effectiveness and reliability [29,30].

Therefore, to achieve consistent results, it is essential to apply tests that include class imbalance corrections (e.g., Gwet’s AC2).

## 5. Conclusions

Our study confirms that PI-FAB scoring is an effective method for achieving consistent evaluations of prostate mpMRI after HIFU. Our results show excellent inter-reader agreement among radiologists with different levels of experience, confirming that the PI-FAB score is highly reproducible. The high number of PI-FAB 1 scores assigned by the most experienced radiologist suggests that HIFU could be a reliable technique in selected patients, but additional validation studies may be necessary to confirm its effectiveness.

## Figures and Tables

**Figure 1 cancers-17-01031-f001:**
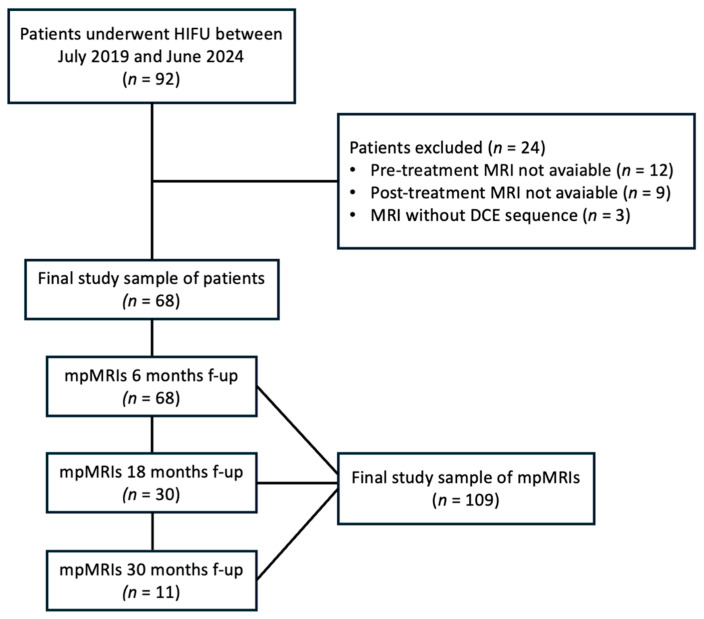
A flowchart of the patient selection process. HIFU, high intensity focused ultrasound; mpMRI, multiparametric magnetic resonance imaging; DCE, dynamic contrast enhanced.

**Figure 2 cancers-17-01031-f002:**
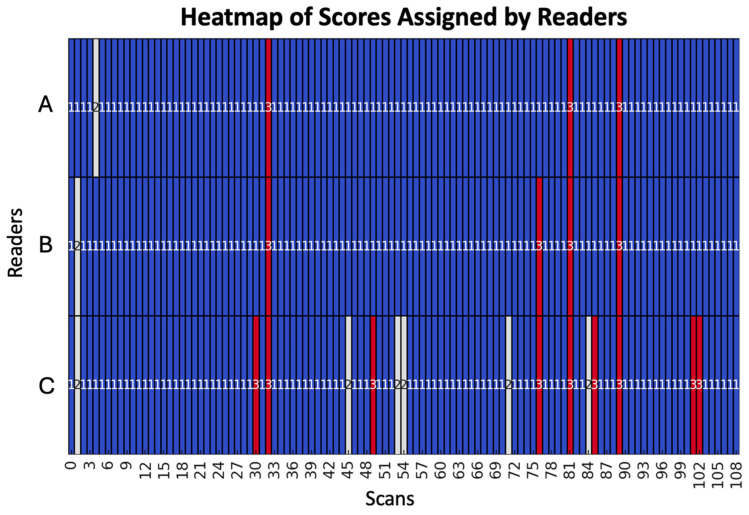
The heatmap illustrates the scores assigned by three different readers (A, B, and C, arranged in descending order of experience) to our series of MRI scans. Each cell in the heatmap corresponds to the score given by a particular reader for a specific scan: 1 (blue), 2 (white), and 3 (red). This visual representation allows for easy comparison of the ratings across readers, highlighting areas of agreement and discrepancy.

**Figure 3 cancers-17-01031-f003:**
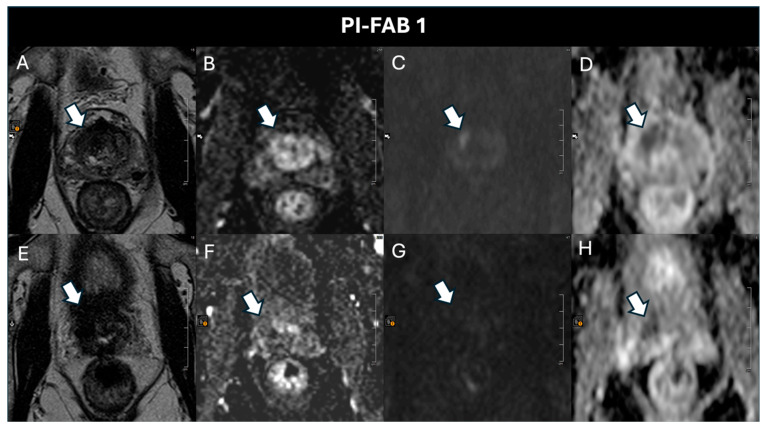
A 75-year-old patient with a pre-treatment PSA of 4.06 ng/mL and a post-treatment PSA of 2.20 ng/mL. The pre-treatment baseline (**A**) T2w, (**B**) DCE, (**C**) high-b-value, and (**D**) ADC map acquisitions show a PI-RADS 4 lesion (arrow) in the right anterior basal-mid transition zone. At histopathology there is a GS 3 + 4 lesion. Post-treatment magnetic resonance imaging after 6 months shows, in the same location, residual fibrosis hypointense on (**E**) T2w and (**H**) the ADC map, without enhancement (**F**) and hyperintensity on the (**G**) high-b-value. The assigned PI-FAB score is, therefore, 1.

**Figure 4 cancers-17-01031-f004:**
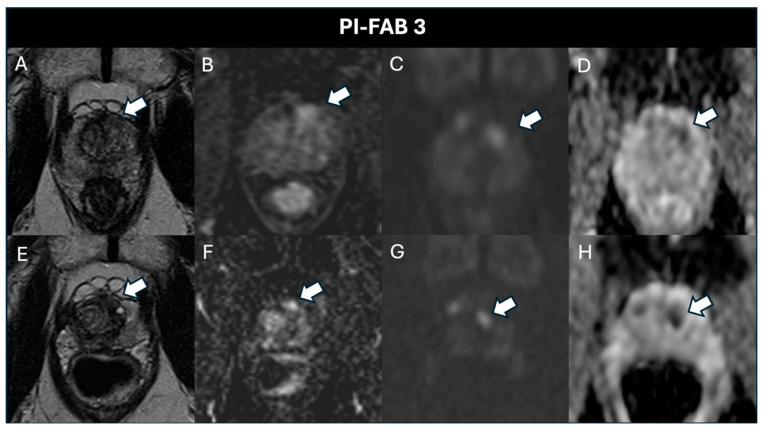
A 60-year-old patient with a pre-treatment PSA of 3.78 ng/mL and a post-treatment PSA of 4.68 ng/mL. The pre-treatment baseline (**A**) T2w, (**B**) DCE, (**C**) high-b-value, and (**D**) ADC map acquisitions show a PI-RADS 4 lesion (arrow) in the left anterior mid transition zone. At histopathology there is a GS 3 + 4 lesion. Post-treatment magnetic resonance imaging after 6 months shows, in the same location, a lesion hypointense on (**E**) T2w and (**H**) the ADC map, with focal enhancement (**F**) and hyperintensity on the (**G**) high-b-value. The assigned PI-FAB score is, therefore, 3.

**Table 1 cancers-17-01031-t001:** The main patient demographics and lesion characteristics.

Variable	Value
Mean age ± SD, y	70.6 ± 8.31
Pre-treatment mean PSA, ng/mL	7.85 ± 1.21
Post-treatment mean PSA, ng/mL	4.64 ± 4.2
Pre-treatment PI-RADS, n (%)	
3	26 (38.2)
4	42 (61.8)
(ISUP/WHO) grade group, n (%)	
1	61 (89.7)
2	7 (10.3)

PSA, prostate-specific antigen; PI-RADS, prostate imaging–reporting and data system; ISUP, International Society of Urological Pathology; WHO, World Health Organization.

**Table 2 cancers-17-01031-t002:** The scores assigned by the most experienced radiologist at 6, 18, and 30 months.

	6 Months	18 Months	30 Months
mpMRI, n	68	30	11
PI-FAB 1, n (%)	64 (94.1)	30 (100)	11 (100)
PI-FAB 2, n (%)	1 (1.5)	0 (0)	0 (0)
PI-FAB 3, n (%)	3 (4.4)	0 (0)	0 (0)

mpMRI, multiparametric magnetic resonance imaging; PI-FAB, Prostate Imaging after Focal Ablation.

## Data Availability

The raw data supporting the conclusions of this article will be made available by the authors on request.

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
