# Peer review of "The Use of PI-FAB Score in Evaluating mpMRI After Focal Ablation of Prostate Cancer: Is It Reliable? Inter-Reader Agreement in a Tertiary Care Referral University Hospital"

_cancers, 2025, doi:10.3390/cancers17061031_

Round 1
Reviewer 1 Report
Comments and Suggestions for Authors
n this paper, the authors evaluated MRI images using the PI-FAB system to achieve high intra-reader agreement following focused ultrasound treatment for prostate cancer. This evaluation method is important because it yields consistent results without relying on subjective interpretations from experienced readers. Although the paper is well written and clear overall, the main takeaway is not immediately evident. Is the paper suggesting that PI-FAB is an effective method for achieving consistent evaluations, or that focused ultrasound is a reliable treatment for cancer? Because the conclusion is unclear, the contribution is difficult to discern.
Additionally, I have a few specific comments:
1. Lines 21–22: It would be helpful to elaborate on what the authors mean by “standardization” and explain the issues that arise in its absence. If the authors collected data using a consistent protocol, then the images should be comparable.
2. Line 24: Change “PIFAB” to “PI-FAB” for consistency.
3. Line 36: Clarify what “PSA” stands for.
4. Line 168: Specify which test was used to calculate the p-value.
5. Lines 283–285: It is unclear whether the high agreement between readers is due to a standardized protocol or if it is inherent to the PI-FAB system.
Reviewer 2 Report
Comments and Suggestions for Authors
It is a retrospective study so the fact should be stated in the abstract. Some excluded patients might have poorer outcome of HIFU than the included patients. The Abstract could add and explain what PI-FAB1 implies.
HIFU treatment was given only to a selected subgroup of patients with localized PCa. So the conclusion should be specified that HIFU is reliable only for this small subgroup of patients with PCa. The period for inclusion of patients is not stated.
The paper could refer to a prospective study of HIFU [1]. The study reported preservation of urinary and sexual function after HIFU. The study used Sexual Health Inventory for Men and InternationalProstate Symptom Score. Such aspects are as important for evaluation of reliability of HIFU as interobserver agreement of mpMRI interpretation between radiologists.
As to algorithm at the hospital 3 patients should have undergone prostate biopsy after the first posttreatment mpMRI. The paper could add whether that happened.
Reviewer 3 Report
Comments and Suggestions for Authors
- In the Simple Summary, “PI-FAB” in line 22, and “mpMRI” in line 23 should be fully spelled out.
- In the Abstract, please include the headings 'Background/Purpose', 'Methods', 'Results', and 'Conclusion' to improve clarity and facilitate understanding of the study.
- In the Abstract, in line 35, please specify the exact years of experience for the “three radiologists with varying levels of experience” to provide more precise information.
- In the Abstract, on line 35, “68 patients” is mentioned again, although “68 consecutive patients” already appears in line 33. Please revise the sentence to avoid redundancy.
- In the Introduction, please add a relevant reference to support the sentence in lines 66- 67.
- In the Materials and Methods, there are too many line breaks.
- In the Results, please add pretreatment tumor size on T2-weighted image, ADC map or contrast enhanced T1-weighted image, and tumor location in the prostate.
- In the Discussion, lines 231-232, please clarify what you are referring to regarding the sensitivity and specificity.
- In the Discussion, lines 237-238, please do not break the line and clarify this example is associated with the previous sentence.
- In the Discussion, lines 254, “So” is not appropriate for an academic paper.
- I recommend elucidating the novel findings of this study, beyond the fact that the number of radiologists differs from previous studies [6,7]. In particular, it seems to me that this study is similar to reference [6]. Therefore, you should discuss the differences between this study and study [6] in more detail. Also, you should clarify why “the use of PI-FAB seems to significantly reduce the differences due to radiologist experience”, as noted in line 251-252.
Overall, the quality of English is good. However, in the Discussion, lines 254, “So” is not appropriate for an academic paper.
Round 2
Reviewer 1 Report
Comments and Suggestions for Authors
The authors did a good job addressing my comments, and I believe the paper should be published.
Comments on the Quality of English LanguageThe language is easy to understand, and the information delivery is precise
Author Response
Comment 1: The authors did a good job addressing my comments, and I believe the paper should be published.
The language is easy to understand, and the information delivery is precise
Response 1: We greatly thank the Reviewer for the constructive comment and for the positive and encouraging feedback regarding our work.
Reviewer 2 Report
Comments and Suggestions for Authors
Thank you for corrective comments to the reviewer 2 comments.
Figure 1 shows patient selection but includes number of mpMRI.
92 patients underwent HIFU for a specified recruitment period, but not the recruitment period for the 68 included patients. If 24 patients were excluded, were the 68 included patients consecutive?
HIFU is a treatment for selected patients with localized PCa. It is not stated in line 21, and line 53.
If some patients were recruited July 2019, follow-up with repeat measurements of PSA might reveal recurrent PCa in nearly 5 years follow-up. The paper stops at 2.5 years of follow-up (line 102).
The authors reported agreement between radiologists, not adverse effects of the treatment. It is a limitation as the authors support HIFU as a reliable and effective treatment (line 315 , 316).
For FAB 3 (at 6 months) a biopsy was recommended (line 150). Was a biopsy carried out? Did it confirm presence of PCa? For patients with FAB 3 at six months, how long were they followed?
Author Response
Thank you for corrective comments to the reviewer 2 comments.
We thank the Reviewer for His/Her work and for the constructive comments.
Figure 1 shows patient selection but includes number of mpMRI.
We thank the reviewer for the comment. Perhaps we did not understand the comment: maybe the Reviewer means that Figure 1 includes both the selected patients and the number of MRIs? Because indeed, that is what we intended to do. The flowchart first explains how many patients were enrolled and then how many MRIs were performed, since our work, being a radiological study, focuses on the number of MRIs performed and subsequently evaluated using PI-FAB
92 patients underwent HIFU for a specified recruitment period, but not the recruitment period for the 68 included patients. If 24 patients were excluded, were the 68 included patients consecutive?
We thank the Reviewer for the valuable comment. Yes, the 68 enrolled patients were consecutive.
HIFU is a treatment for selected patients with localized PCa. It is not stated in line 21, and line 53.
We thank the Reviewer for the valuable comment. We had already made this change in the main text in Revision Round 1 (line 315-316); we had not included it in the abstract due to length constraints. However, we have now also modified the abstract accordingly (line 53).
If some patients were recruited July 2019, follow-up with repeat measurements of PSA might reveal recurrent PCa in nearly 5 years follow-up. The paper stops at 2.5 years of follow-up (line 102).
We appreciate the Reviewer’s observation. Initially, very few patients were recruited because few HIFU procedures were performed; the volume of examinations has increased over the years, and therefore patient recruitment has also increased. Regarding recurrences, as we stated, we had 3 recurrences which were also accompanied by an increase in PSA levels.
The authors reported agreement between radiologists, not adverse effects of the treatment. It is a limitation as the authors support HIFU as a reliable and effective treatment (line 315 , 316).
We thank the Reviewer for the observation, but our work is radiological, not clinical, study focused on evaluating a score for assessing MRIs after focal therapy. The adverse effects of HIFU are outside the scope of our work. Furthermore, we have suggested that HIFU may be an interesting treatment in selected patients because in subsequent MRI follow-ups, very few patients show disease recurrence; we do not make considerations regarding the safety of the method or its adverse effects.
For FAB 3 (at 6 months) a biopsy was recommended (line 150). Was a biopsy carried out? Did it confirm presence of PCa? For patients with FAB 3 at six months, how long were they followed?
We thank the Reviewer for the clarification, but perhaps the Reviewer overlooked the fact that we had specified (line 200-202) “At the retrospective analysis of the post-biopsy histopatological reports of the 3 patients with a PI-FAB score of 3 reported by the most experienced radiologist, they were all confirmed as PCa WHO/ISUP grade 2.” Patients with FAB 3 at 6 months were not followed further but were biopsed and referred to the urologist who decided the subsequent treatment.
Reviewer 3 Report
Comments and Suggestions for Authors
The authors have satisfactorily addressed all my concerns.
Author Response
Comment 1: The authors have satisfactorily addressed all my concerns.
Response 1: We greatly thank the Reviewer for the constructive and positive feedback.